# CADet: Fully Self-Supervised Anomaly Detection With Contrastive Learning

## Abstract

Handling out-of-distribution (OOD) samples has become a major stake in the real-world deployment of machine learning systems. This work explores the application of self-supervised contrastive learning to the simultaneous detection of two types of OOD samples: unseen classes and adversarial perturbations. Since in practice the distribution of such samples is not known in advance, we do not assume access to OOD examples. We show that similarity functions trained with contrastive learning can be leveraged with the maximum mean discrepancy (MMD) two-sample test to verify whether two independent sets of samples are drawn from the same distribution. Inspired by this approach, we introduce CADet (Contrastive Anomaly Detection), a method based on image augmentations to perform anomaly detection on single samples. CADet compares favorably to adversarial detection methods to detect adversarially perturbed samples on ImageNet. Simultaneously, it achieves comparable performance to unseen label detection methods on two challenging benchmarks: ImageNet-O and iNaturalist. CADet is fully self-supervised and requires neither labels for in-distribution samples nor access to OOD examples.

## 1 Introduction

While modern machine learning systems have achieved countless successful real-world applications, handling out-of-distribution (OOD) inputs remains a tough challenge of significant importance. The problem is especially acute for high-dimensional problems like image classification. Models are typically trained in a close-world setting but inevitably faced with novel input classes when deployed in the real world. The impact can range from displeasing customer experience to dire consequences in the case of safety-critical applications such as autonomous driving (Kitt et al., 2010) or medical analysis (Schlegl et al., 2017a). Although achieving high accuracy against all meaningful distributional shifts is the most desirable solution, it is particularly challenging. An efficient method to mitigate the consequences of unexpected inputs is to perform anomaly detection, which allows the system to anticipate its inability to process unusual inputs and react adequately.

Anomaly detection methods generally rely on one of three types of statistics: features, logits, and softmax probabilities, with some systems leveraging a mix of these (Wang et al., 2022). An anomaly score $f(x)$ is computed, and then detection with threshold $\tau$ is performed based on whether $f(x) > \tau$. The goal of a detection system is to find an anomaly score that efficiently discriminates between in-distribution and out-of-distribution samples. However, the common problem of these systems is that different distributional shifts will unpredictably affect these statistics. Accordingly, detection systems either achieve good performance on specific types of distributions or require tuning on OOD samples. In both cases, their practical use is severely limited. Motivated by these issues, recent work has tackled the challenge of designing detection systems for unseen classes without prior knowledge of the unseen label set or access to OOD samples (Winkens et al., 2020; Tack et al., 2020; Wang et al., 2022).

We first investigate the use of maximum mean discrepancy two-sample test (MMD) (Gretton et al., 2012) in conjunction with self-supervised contrastive learning to assess whether two sets of samples have been drawn from the same distribution. Motivated by the strong testing power of this method, we then introduce a statistic inspired by MMD and leveraging contrastive transformations. Based on this statistic, we propose CADet (Contrastive Anomaly Detection), which is able to detect OOD samples

from single inputs and performs well on both label-based and adversarial detection benchmarks, without requiring access to any OOD samples to train or tune the method.

Only a few works have addressed these tasks simultaneously. These works either focus on particular in-distribution data such as medical imaging for specific diseases (Uwimana1 & Senanayake, 2021) or evaluate their performances on datasets with very distant classes such as CIFAR10 (Krizhevsky, 2009), SVHN (Netzer et al., 2011), and LSUN (Yu et al., 2015), resulting in simple benchmarks that do not translate to general real world applications (Lee et al., 2018).

**Contributions:** Our main contributions are as follows:

- We use similarity functions learned by self-supervised contrastive learning with MMD to show that the test sets of CIFAR10 and CIFAR10.1 (Recht et al., 2019) have different distributions.
- We propose a novel improvement to MMD and show it can also be used to confidently detect distributional shifts when given a small number of samples.
- We introduce CADet, a fully self-supervised method for anomaly detection, and show it outperforms current methods in adversarial detection tasks while performing well on class-based OOD detection.

The outline is as follows: in Section 2, we discuss relevant previous work. Section 3 describes the self-supervised contrastive method based on SimCLRv2 (Chen et al., 2020b) used in this work. Section 4 explores the application of learned similarity functions in conjunction with MMD to verify whether two independent sets of samples are drawn from the same distribution. Section 5 presents CADet and evaluates its empirical performance. Finally, we discuss results and limitations in Section 6.

## 2 RELATED WORK

We propose a self-supervised contrastive method for anomaly detection (both unknown classes and adversarial attacks) inspired by MMD. Thus, our work intersects with the MMD, label-based OOD detection, adversarial detection, and self-supervised contrastive learning literature.

**MMD** two-sample test has been extensively studied (Gretton et al., 2012; Wenliang et al., 2019; Gretton et al., 2009; Sutherland et al., 2016; Chwialkowski et al., 2015; Jitkrittum et al., 2016), though it is, to the best of our knowledge, the first time a similarity function trained via contrastive learning is used in conjunction with MMD. Liu et al. (2020a) uses MMD with a deep kernel trained on a fraction of the samples to argue that CIFAR10 and CIFAR10.1 have different test distributions. We build upon that work by confirming their finding with higher confidence levels, using fewer samples.

**Label-based OOD detection methods** discriminate samples that differ from those in the training distribution. We focus on unsupervised OOD detection in this work, *i.e.*, we do not assume access to data labeled as OOD. Unsupervised OOD detection methods include density-based (Zhai et al., 2016; Nalisnick et al., 2018; 2019; Choi et al., 2018; Du & Mordatch, 2019; Ren et al., 2019; Serrà et al., 2019; Grathwohl et al., 2019; Liu et al., 2020b; Dinh et al., 2016), reconstruction-based (Schlegl et al., 2017b; Zong et al., 2018; Deecke et al., 2018; Pidhorskyi et al., 2018; Perera et al., 2019; Choi et al., 2018), one-class classifiers (Schölkopf et al., 1999; Ruff et al., 2018), self-supervised (Golan & El-Yaniv, 2018; Hendrycks et al., 2019b; Bergman & Hoshen, 2020; Tack et al., 2020), and supervised approaches (Liang et al., 2017; Hendrycks & Gimpel, 2016), though some works do not fall into any of these categories (Wang et al., 2022).

**Adversarial detection** discriminates adversarial samples from the original data. Adversarial samples are generated by minimally perturbing actual samples to produce a change in the model's output, such as a misclassification. Most works rely on the knowledge of some attacks for training (Abusnaina et al., 2021; Metzen et al., 2017; Feinman et al., 2017; Lust & Condurache, 2020; Zuo & Zeng, 2021; Papernot & McDaniel, 2018; Ma et al., 2018), with the exception of Hu et al. (2019).

**Self-supervised contrastive learning** methods (Wu et al., 2018; He et al., 2020; Chen et al., 2020a;b) are commonly used to pre-train a model from unlabeled data to solve a downstream task such as image classification. Contrastive learning relies on instance discrimination trained with a contrastive loss (Hadsell et al., 2006) such as infoNCE (Gutmann & Hyvärinen, 2010).

**Contrastive learning for OOD detection** aims to find good representations for detecting OOD samples in a supervised (Liu & Abbeel, 2020; Khalid et al., 2022) or unsupervised (Winkens et al., 2020; Mohseni et al., 2020; Sehwag et al., 2021) setting. Perhaps the closest work in the literature is CSI (Tack et al., 2020), which found SimCLR features to have good discriminative power for unknown classes detection and leveraged similarities between transformed samples in their score. However, this method is not well-suited for adversarial detection. CSI ignores the similarities between different transformations of a same sample, an essential component to perform adversarial detection (see Section 6.2). In addition, CSI scales their score with the norm of input representations. While efficient on samples with unknown classes, it is unreliable on adversarial perturbations, which typically increase representation norms.

## 3 Contrastive model

We build our model on top of SimCLRv2 (Chen et al., 2020b) for its simplicity and efficiency. It is composed of an encoder backbone network $f_\theta$ as well as a 3-layer contrastive head $h_{\theta'}$. Given an in-distribution sample $\mathcal{X}$, a similarity function *sim*, and a distribution of training transformations $\mathcal{T}_{train}$, the goal is to simultaneously maximize $\mathbb{E}_{x \sim \mathcal{X}; t_0, t_1 \sim \mathcal{T}_{train}} [sim(h_{\theta'} \circ f_\theta(t_0(x)), h_{\theta'} \circ f_\theta(t_1(x)))]$ and minimize $\mathbb{E}_{x,y \sim \mathcal{X}; t_0, t_1 \sim \mathcal{T}_{train}} [sim(h_{\theta'} \circ f_\theta(t_0(x)), h_{\theta'} \circ f_\theta(t_1(y)))]$, i.e., we want to learn representations in which random transformations of a same example are close while random transformations of different examples are distant.

To achieve this, given an input batch $\{x_i\}_{i=1,...,N}$, we compute the set $\{x_i^{(j)}\}_{j=0,1; i=1,...,N}$ by applying two transformations independently sampled from $\mathcal{T}_{train}$ to each $x_i$. We then compute the embeddings $z_i^{(j)} = h_{\theta'} \circ f_\theta(x_i^{(j)})$ and apply the following contrastive loss:

$$L(\mathbf{z}) = \sum_{i=1,...,N} -\log \frac{e^{sim(z_i^{(0)}, z_i^{(1)})/\tau}}{\sum_{j \in \{1,...,N\}} e^{sim(z_i^{(0)}, z_j^{(1)})/\tau} + \sum_{j \in \{1,...,N\} \setminus i} e^{sim(z_i^{(0)}, z_j^{(0)})/\tau}}, \quad (1)$$

where $\tau$ is the temperature hyperparameter and $sim(x,y) = \frac{\langle x | y \rangle}{\|x\|_2 \|y\|_2}$ is the *cosine* similarity.

**Hyperparameters:** We follow as closely as possible the setting from SimCLRv2 with a few modifications to adapt to hardware limitations. In particular, we use the LARS optimizer (You et al., 2017) with learning rate 1.2, momentum 0.9, and weight decay $10^{-4}$. Iteration-wise, we scale up the learning rate for the first 40 epochs linearly, then use an iteration-wise cosine decaying schedule until epoch 800, with temperature $\tau = 0.1$. We train on 8 $V100$ GPUs with an accumulated batch size of 1024. We compute the contrastive loss on all batch samples by aggregating the embeddings computed by each GPU. We use synchronized BatchNorm and fp32 precision and do not use a memory buffer. We use the same set of transformations, i.e., Gaussian blur and horizontal flip with probability 0.5, color jittering with probability 0.8, random crop with scale uniformly sampled in $[0.08, 1]$, and grayscale with probability 0.2.

For computational simplicity and comparison with previous work, we use a ResNet50 encoder architecture with final features of size 2048. Following SimCLRv2, we use a three-layer fully connected contrastive head with hidden layers of width 2048 using ReLU activation and batchNorm and set the last layer projection to dimension 128. For evaluation, we use the features produced by the encoder without the contrastive head. We do not, at any point, use supervised fine-tuning.

## 4 MMD two-sample test

The **Maximum Mean Discrepancy (MMD)** is a statistic used in the MMD two-sample test to assess whether two sets of samples $S_{\mathbb{P}}$ and $S_{\mathbb{Q}}$ are drawn from the same distribution. It estimates the expected difference between the intra-set distances and the across-sets distances.

**Definition 4.1** (Gretton et al. (2012)). Let $k : \mathcal{X} \times \mathcal{X} \to \mathbb{R}$ be the kernel of a reproducing Hilbert space $\mathcal{H}_k$, with feature maps $k(\cdot, x) \in \mathcal{H}_k$. Let $X, X' \sim \mathbb{P}$ and $Y, Y' \sim \mathbb{Q}$. Under mild integrability

conditions,

$$MMD(\mathbb{P}, \mathbb{Q}; \mathcal{H}_k) := \sup_{f \in \mathcal{H}, \|f\|_{\mathcal{H}_k} \leq 1} |\mathbb{E}[f(X)] - \mathbb{E}[f(Y)]| \tag{2}$$

$$= \sqrt{\mathbb{E}[k(X, X') + k(Y, Y') - 2k(X, Y)]}. \tag{3}$$

Given two sets of $n$ samples $S_{\mathbb{P}} = \{X_i\}_{i \leq n}$ and $S_{\mathbb{Q}} = \{Y_i\}_{i \leq n}$, respectively drawn from $\mathbb{P}$ and $\mathbb{Q}$, we can compute the following unbiased estimator Liu et al. (2020a):

$$\widehat{MMD}_u^2(S_{\mathbb{P}}, S_{\mathbb{Q}}; k) := \frac{1}{n(n-1)} \sum_{i \neq j} (k(X_i, X_j) + k(Y_i, Y_j) - k(X_i, Y_j) - k(Y_i, X_j)). \tag{4}$$

Under the null hypothesis $\mathfrak{h}_0 : \mathbb{P} = \mathbb{Q}$, this estimator follows a normal distribution of mean 0 (Gretton et al., 2012). Its variance can be directly estimated (Gretton et al., 2009), but it is simpler to perform a permutation test as suggested in Sutherland et al. (2016), which directly yields a $p$-value for $\mathfrak{h}_0$. The idea is to use random splits $X, Y$ of the input sample sets to obtain $n_{perm}$ different (though not independent) samplings of $\widehat{MMD}_u^2(X, Y; k)$, which approximate the distribution of $\widehat{MMD}_u^2(S_{\mathbb{P}}, S_{\mathbb{P}}; k)$ under the null hypothesis.

Liu et al. (2020a) train a deep kernel to maximize the test power of the MMD two-sample test on a training split of the sets of samples to test. We propose instead to use our learned similarity function without any fine-tuning. Note that we return the $p$-value $\frac{1}{n_{perm}+1}\left(1 + \sum_{i=1}^{n_{perm}} \mathbb{1}(p_i \geq est)\right)$ instead of $\frac{1}{n_{perm}} \sum_{i=1}^{n_{perm}} \mathbb{1}(p_i \geq est)$. Indeed, under the null hypothesis $\mathbb{P} = \mathbb{Q}$, $est$ and $p_i$ are drawn from the same distribution, so for $j \in \{0, 1, \ldots, n_{perm}\}$, the probability for $est$ to be smaller than exactly $j$ elements of $\{p_i\}$ is $\frac{1}{n_{perm}+1}$. Therefore, the probability that $j$ elements or less of $\{p_i\}_i$ are larger than $est$ is $\sum_{i=0}^{j} \frac{1}{n_{perm}+1} = \frac{j+1}{n_{perm}+1}$. While this change has a small impact for large values of $n_{perm}$, it is essential to guarantee that we indeed return a correct $p$-value. Notably, the algorithm of Liu et al. (2020a) has a probability $\frac{1}{n_{perm}} > 0$ to return an output of 0.00 even under the null hypothesis.

Additionnally, we propose an improvement of MMD called MMD-CC (MMD with Clean Calibration). Instead of computing $p_i$ based on random splits of $S_{\mathbb{P}} \bigcup S_{\mathbb{Q}}$, we require as input two disjoint sets of samples drawn from $\mathbb{P}$ and compute $p_i$ based on random splits of $S_{\mathbb{P}}^{(1)} \bigcup S_{\mathbb{P}}^{(2)}$ (see Algorithm 1). This change requires to use twice as many samples from $\mathbb{P}$, but reduces the variance induced by the random splits of $S_{\mathbb{P}} \bigcup S_{\mathbb{Q}}$, which is significant when the number of samples is small.

## 4.1 DISTRIBUTION SHIFT BETWEEN CIFAR-10 AND CIFAR-10.1 TEST SETS

After years of evaluation of popular supervised architectures on the test set of CIFAR-10 (Krizhevsky, 2009), modern models may overfit it through their hyperparameter tuning and structural choices. CIFAR-10.1 (Recht et al., 2019) was collected to verify the performances of these models on a *truly* independent sample from the training distribution. The authors note a consistent drop in accuracy across models and suggest it could be due to a distributional shift, though they could not demonstrate it. Recent work (Liu et al., 2020a) leveraged the two-sample test to provide strong evidence of distributional shifts between the test sets of CIFAR-10 and CIFAR-10.1. We run MMD-

---

**Algorithm 1** MMD-CC two-sample test

**Input:** $S_{\mathbb{P}}^{(1)}, S_{\mathbb{P}}^{(2)}, S_{\mathbb{Q}}, n_{perm}, sim$

$\quad est \leftarrow \widehat{MMD}_u^2(S_{\mathbb{P}}^{(1)}, S_{\mathbb{Q}}; sim)$

$\quad$ **for** $i = 1, 2, \ldots, n_{perm}$ **do**

$\quad\quad$ Randomly split $S_{\mathbb{P}}^{(1)} \bigcup S_{\mathbb{P}}^{(2)}$ into two disjoint sets $X, Y$ of equal size

$\quad\quad p_i \leftarrow \widehat{MMD}_u^2(X, Y; sim)$

$\quad$ **end for**

**Output:** $p$: $\frac{1}{1+n_{perm}}\left(1 + \sum_{i=1}^{n_{perm}} \mathbb{1}(p_i \geq est)\right)$

---

CC and MMD two-sample tests for 100 different samplings of $S_{\mathbb{P}}^{(1)}, S_{\mathbb{P}}^{(2)}, S_{\mathbb{Q}}$, using every time $n_{perm} = 500$, and rejecting $\mathfrak{h}_0$ when the obtained $p$-value is below the threshold $\alpha = 0.05$. We also report results using cosine similarity applied to the features of supervised models as a comparative baseline. We report the results in Table 1 for a range of sample sizes. We compare the results to three

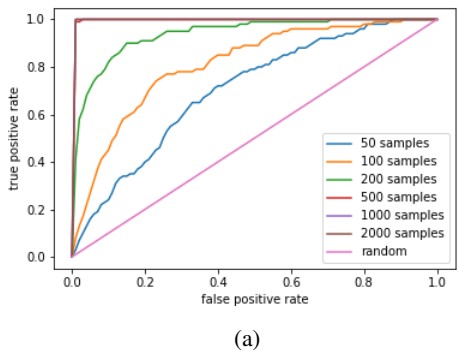 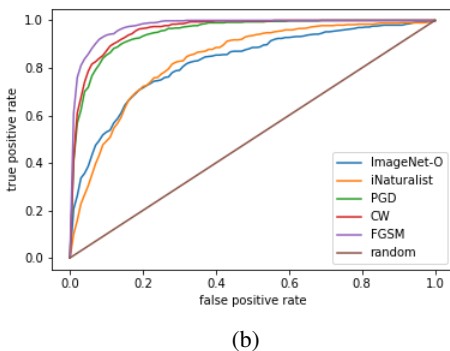

(a)            (b)

Figure 1: ROC curves for MMD-CC two-sample test on CIFAR10 vs CIFAR10.1 using different sample sizes (left) and for CADet anomaly detection on different out-distributions (right).

Table 1: Average rejection rates of $\mathfrak{h}_0$ on CIFAR-10 vs CIFAR-10.1 for $\alpha = 0.05$ across different sample sizes $n$, using a ResNet50 backbone.

|  | n=2000 | n=1000 | n=500 | n=200 | n=100 | n=50 |
|---|---|---|---|---|---|---|
| ME∼Chwialkowski et al. (2015) | 0.588 | - | - | - | - | - |
| C2ST-L∼Cheng & Cloninger (2019) | 0.529 | - | - | - | - | - |
| MMD-D∼Liu et al. (2020a) | 0.744 | - | - | - | - | - |
| MMD + SimCLRv2 (ours) | **1.00** | **1.00** | **0.997** | **0.702** | **0.325** | **0.154** |
| MMD-CC + SimCLRv2 (ours) | **1.00** | **1.00** | **0.997** | 0.686 | 0.304 | 0.150 |
| MMD + Supervised (ours) | **1.00** | **1.00** | 0.884 | 0.305 | 0.135 | 0.103 |
| MMD-CC + Supervised (ours) | **1.00** | **1.00** | 0.870 | 0.298 | 0.131 | 0.096 |

competitive methods reported in Liu et al. (2020a): Mean embedding (ME) (Chwialkowski et al., 2015; Jitkrittum et al., 2016), MMD-D (Liu et al., 2020a), and C2ST-L (Cheng & Cloninger, 2019). Finally, we show in Figure 1a the ROC curves of the proposed model for different sample sizes.

Other methods in the literature do not use external data for pre-training, as we do with ImageNet, which makes a fair comparison difficult. However, it is noteworthy that our learned similarity can very confidently distinguish samples from the two datasets, even in settings with fewer samples available. Furthermore, while we achieve excellent results even with a supervised network, our model trained with contrastive learning outperforms the supervised alternative very significantly. We note however that with such high number of samples available, MMD-CC performs slightly worse than MMD. Finally, we believe the confidence obtained with our method decisively concludes that CIFAR10 and CIFAR10.1 have different distributions, which is likely the primary explanation for the significant drop in performances across models on CIFAR10.1, as conjectured by Recht et al. (2019). The difference in distribution between CIFAR10 and CIFAR10.1 is neither based on label set nor adversarial perturbations, making it an interesting task.

### 4.2 DETECTION OF DISTRIBUTIONAL SHIFTS FROM SMALL NUMBER OF SAMPLES

Given a small set of samples with potential unknown classes or adversarial attacks, we can similarly use the two-sample test with our similarity function to verify whether these samples are in-distribution (Gao et al., 2021). In particular, we test for samples drawn from ImageNet-O, iNaturalist, and PGD perturbations, with sample sizes ranging from 3 to 20. For these experiments, we sample $S_{\mathbb{P}}^{(1)}$ and $S_{\mathbb{P}}^{(2)}$ 5000 times across all of ImageNet's validation set and compare their MMD and MMD-CC estimators to the one obtained from $S_{\mathbb{P}}$ and $S_{\mathbb{Q}}$. We report in Table 2 the AUROC of the resulting detection and compare it to the ones obtained with a supervised ResNet50 as the baseline.

Such a setting where we use several samples assumed to be drawn from a same distribution to perform detection is uncommon, and we are not aware of prior baselines in the literature. Despite using very

Table 2: AUROC for detection using two-sample test on 3 to 20 samples drawn from ImageNet and from ImageNet-O, iNaturalist or PGD perturbations, with a ResNet50 backbone.

| | ImageNet-O | | | | iNaturalist | | | | PGD | | | |
|---|---|---|---|---|---|---|---|---|---|---|---|---|
| n_samples | 3 | 5 | 10 | 20 | 3 | 5 | 10 | 20 | 3 | 5 | 10 | 20 |
| MMD + SimCLRv2 | 64.3 | 72.4 | 86.9 | 97.6 | 88.3 | 97.6 | 99.5 | 99.5 | 35.2 | 53.8 | 86.6 | 98.8 |
| MMD-CC + SimCLRv2 | **65.3** | **73.2** | **88.0** | **97.7** | 95.4 | 99.2 | 99.5 | 99.5 | **70.5** | **84.0** | **96.6** | **99.5** |
| MMD + Supervised | 62.7 | 69.7 | 83.2 | 96.4 | 91.8 | 98.7 | 99.5 | 99.5 | 20.0 | 22.5 | 33.0 | 57.5 |
| MMD-CC + Supervised | 62.6 | 71.0 | 85.5 | 97.2 | **98.0** | **99.5** | 99.5 | 99.5 | 57.4 | 61.3 | 70.5 | 85.8 |

few samples ($3 \leq n \leq 20$), our method can detect OOD samples with high confidence. We observe particularly outstanding performances on iNaturalist, which is easily explained by the fact that the subset we are using (cf. Section 1) only contains plant species, logically inducing an abnormally high similarity within its samples. Furthermore, we observe that MMD-CC performs significantly better than MMD, especially on detecting samples perturbed by PGD.

Although our method attains excellent detection rates for sufficient numbers of samples, the requirement to have a set of samples all drawn from the same distribution to perform the test makes it unpractical for real-world applications. In the following section, we present CADet, a detection method inspired by MMD but applicable to anomaly detection with single inputs.

## 5 CADET: CONTRASTIVE ANOMALY DETECTION

While the numbers in Section 4 demonstrate the reliability of two-sample test coupled with contrastive learning for identifying distributional shifts, it requires several samples from the same distribution, which is generally unrealistic for practical detection purposes. This section presents CADet, a method to leverage contrastive learning for anomaly detection on single samples from OOD distributions.

Self-supervised contrastive learning trains a similarity function $s$ to maximize the similarity between augmentations of the same sample, and minimize the similarity between augmentations of different samples. Given an input sample $x^{test}$, we propose to leverage this property to perform anomaly detection on $x^{test}$, taking inspiration from MMD two-sample test. More precisely, given a transformation distribution $\mathcal{T}_{val}$, we compute $n_{trs}$ random transformations $x_i^{test}$ of $x_{test}$, as well as $n_{trs}$ random transformations $x_i^{(k)}$ on each sample $x^{(k)}$ of a held-out validation dataset $X_{val}^{(1)}$. We then compute the intra-similarity and out-similarity:

$$m^{in}(x^{test}) := \frac{\sum\limits_{i \neq j} s(x_i^{test}, x_j^{test})}{n_{trs}(n_{trs} + 1)}, \qquad m^{out}(x^{test}) := \frac{\sum\limits_{x^{(k)} \in X_{val}^{(1)}} \sum\limits_{i,j} s(x_i^{test}, x_j^{(k)})}{n_{trs}^2 \times |X_{val}^{(1)}|}. \qquad (5)$$

We finally define the following statistic to perform detection:

$$score_C := m^{in} + \gamma m^{out}. \qquad (6)$$

**Calibration:** since we do not assume knowledge of OOD samples, it is difficult *a priori* to tune $\gamma$, although crucial to balance information between intra-sample similarity and cross-sample similarity. As a workaround, we calibrate $\gamma$ by equalizing the variance between $m^{in}$ and $\gamma m^{out}$ on a second set of validation samples $X_{val}^{(2)}$:

$$\gamma = \sqrt{\frac{Var\{m^{in}(x), x \in X_{val}^{(2)}\}}{Var\{m^{out}(x), x \in X_{val}^{(2)}\}}}. \qquad (7)$$

Rather than evaluating the false positive rate (FPR) for a range of possible thresholds $\tau$, we use the hypothesis testing approach to compute the p-value:

$$p_{value}(x^{test}) = \frac{|\{x \in X_{val}^{(2)} \quad s.t. \quad score_C(x) < score_C(x^{test})\}| + 1}{|\{X_{val}^{(2)}\}| + 1}. \qquad (8)$$

Algorithm 2 and Algorithm 3 detail the calibration and the testing steps, respectively. Setting a threshold $p \in [0,1]$ for the $p_{value}$ will result in a FPR of mean $p$, with a variance dependant of $|X_{val}^{(2)}|$. Section 5.1 further describes our experimental setting.

---

**Algorithm 2** CADet calibration step

---

**Input:** $X_{val}^{(1)}$, $X_{val}^{(2)}$, $\mathcal{T}_{eval}$, learned similarity function $s$, various hyperparameters used below;

1: **for** $x^{(1)} \in X_{val}^{(1)}$ **do**
2:     **for** $i = 1, 2, \ldots, n_{trs}$ **do**
3:         Sample $t$ from $\mathcal{T}_{eval}$
4:         $x_i^{(1)} \leftarrow t(x^{(1)})$
5:     **end for**
6: **end for**
7: $k \leftarrow 0$
8: **for** $x^{(2)} \in X_{val}^{(2)}$ **do**
9:     **for** $i = 1, 2, \ldots, n_{trs}$ **do**
10:        Sample $t$ from $\mathcal{T}_{eval}$
11:        $x_i^{(2)} \leftarrow t(x^{(2)})$
12:     **end for**
13:     $m_k^{in} \leftarrow \sum\limits_{i \neq j}^{n_{trs}} s(x_i^{(2)}, x_j^{(2)})$
14:     $m_k^{out} \leftarrow \sum\limits_{x^{(1)} \in X_{val}^{(1)}} \sum\limits_{i,j}^{n_{trs}} s(x_i^{(1)}, x_j^{(2)})$
15:     $k \leftarrow k + 1$
16: **end for**
17: $m^{in} \leftarrow \frac{m^{in}}{n_{trs}(n_{trs}+1)}$
18: $m^{out} \leftarrow \frac{m^{out}}{n_{trs}^2 \#\{X_{val}^{(1)}\}}$
19: $V_{in}, V_{out} = Var(m^{in}), Var(m^{out})$
20: $\gamma \leftarrow \sqrt{\frac{V_{in}}{V_{out}}}$
21: $k \leftarrow 0$
22: **for** $x^{(2)} \in X_{val}^{(2)}$ **do**
23:     $score_k \leftarrow m_k^{in} + \gamma \times m_k^{out}$
24:     $k \leftarrow k + 1$
25: **end for**
    **Output:** coefficient: $\gamma$, scores: $score_k$, transformed samples: $x_i^{(1)}$

---

**Algorithm 3** CADet testing step

---

**Input:** transformed samples: $x_i^{(1)}$, scores: $scores$, test sample: $x^{test}$, coefficient: $\gamma$, trasnformation set: $\mathcal{T}_{eval}$;

1: **for** $i = 1, 2, \ldots, n_{trs}$ **do**
2:     Sample $t$ from $\mathcal{T}_{eval}$
3:     $x_i^{test} \leftarrow t(x^{test})$
4: **end for**
5: $m^{in} \leftarrow \sum\limits_{i \neq j}^{n_{trs}} s(x_i^{test}, x_j^{test})$
6: $m^{out} \leftarrow \sum\limits_{x^{(1)} \in X_{val}^{(1)}} \sum\limits_{i,j}^{n_{trs}} s(x_i^{test}, x_j^{(1)})$
7: $score^{te} \leftarrow \frac{m^{in}}{n_{trs}(n_{trs}+1)} + \gamma \frac{m^{out}}{n_{trs}^2 \#\{X_{val}^{(1)}\}}$
8: $rank \leftarrow 0$
9: **for** $score^{val} \in scores$ **do**
10:     **if** $score^{val} < score^{te}$ **then**
11:        $rank \leftarrow rank + 1$
12:     **end if**
13: **end for**
14: $p_{val} \leftarrow \frac{rank+1}{\#\{scores\}+1}$
    **Output:** p-value: $p_{val}$

---

## 5.1 EXPERIMENTS

For all evaluations, we use the same transformations as SimCLRv2 except color jittering, Gaussian blur and grayscaling. We fix the random crop scale to $0.75$. We use $|\{X_{val}^{(2)}\}| = 2000$ in-distribution samples, $|\{X_{val}^{(1)}\}| = 300$ separate samples to compute cross-similarities, and 50 transformations per sample. We pre-train a ResNet50 with ImageNet as in-distribution.

**Unknown classes detection:** we use two challenging benchmarks for the detection of unknown classes. iNaturalist using the subset in Huang & Li (2021) made of plants with classes that do not intersect ImageNet. Wang et al. (2022) noted that this dataset is particularly challenging due to proximity of its classes. We also evaluate on ImageNet-O (Hendrycks et al., 2021); explicitly designed to be challenging for OOD detection with ImageNet as in-distribution. We compare to recent works and report the AUROC scores in Table 3.

**Adversarial detection:** for adversarial detection, we generate adversarial attacks on the validation partition of ImageNet against a pre-trained ResNet50 using three popular attacks: PGD (Madry et al.,

Table 3: AUROC for OOD detection on ImageNet-O and iNaturalist with ResNet50 backbone.

| | Training | iNaturalist | ImageNet-O | Average |
|---|---|---|---|---|
| MSP Hendrycks & Gimpel (2016) | | 88.58 | 56.13 | 72.36 |
| Energy Liu et al. (2020c) | | 80.50 | 53.95 | 67.23 |
| ODIN Liang et al. (2017) | | 86.48 | 52.87 | 69.68 |
| MaxLogit Hendrycks et al. (2019a) | | 86.42 | 54.39 | 70.41 |
| KL Matching Hendrycks et al. (2019a) | Supervised | 90.48 | 67.00 | 78.74 |
| ReAct Sun et al. (2021) | | 87.27 | 68.02 | 77.65 |
| Mahalanobis Lee et al. (2018) | | 89.48 | 80.15 | 84.82 |
| Residual Wang et al. (2022) | | 84.63 | 81.15 | 82.89 |
| ViM Wang et al. (2022) | | 89.26 | 81.02 | **85.14** |
| **CADet (ours)** | Supervised | **95.28** | 70.73 | 83.01 |
| | Self-supervised (contrastive) | 83.42 | **82.29** | 82.86 |

Table 4: AUROC for adversarial detection on ImageNet against PGD, CW and FGSM attacks, with ResNet50 backbone.

| | Tuned on Adv | Training | PGD | CW | FGSM | Average |
|---|---|---|---|---|---|---|
| ODIN (Liang et al., 2017) | Yes | Supervised | 62.30 | 60.29 | 68.10 | 63.56 |
| | | Contrastive | 59.91 | 60.23 | 64.99 | 61.71 |
| Hu (Hu et al., 2019) | Yes | Supervised | 84.31 | 84.29 | 77.95 | 82.18 |
| | | Contrastive | 94.80 | 95.19 | 78.18 | 89.39 |
| Hu (Hu et al., 2019) + self-calibration | No | Supervised | 66.40 | 59.58 | 71.02 | 65.67 |
| | | Contrastive | 75.69 | 75.74 | 69.20 | 73.54 |
| **CADet (ours)** | No | Supervised | 75.25 | 71.02 | 83.45 | 76.57 |
| | | Contrastive | **94.88** | **95.93** | **97.56** | **96.12** |

2017), CW (Carlini & Wagner, 2017), and FGSM (Goodfellow et al., 2014). We follow the tuning suggested by Abusnaina et al. (2021), i.e. PGD: norm $L_\infty$, $\delta = 0.02$, step size 0.002, 50 iterations; CW: norm $L_2$, $\delta = 0.10$, learning rate of 0.03, and 50 iterations; FGSM: norm $L_\infty$, $\delta = 0.05$. We compare our results with ODIN (Liang et al., 2017), which achieves good performances in Lee et al. (2018) despite not being designed for adversarial detection, and to Hu et al. (2019). Most other existing adversarial detection methods assume access to adversarial samples during training (see Section 2). While both of these works use adversarial samples to tune hyperparameters, we additionally propose a modification to Hu et al. (2019) to perform auto-calibration based on the mean and variance of the criterions on clean data, similarly to CADet's calibration step. We report the AUROC scores in Table 4 and illustrate them with ROC curves against each anomaly type in Figure 1b.

## 6 DISCUSSION

### 6.1 RESULTS

CADet performs particularly well on adversarial detection, surpassing alternatives by a wide margin. We argue that self-supervised contrastive learning is a suitable mechanism for detecting classification attacks due to its inherent label-agnostic nature. Interestingly, Hu et al. (2019) also benefits from contrastive pre-training, achieving much higher performances than with a supervised backbone. However, it is very reliant on calibrating on adversarial samples, since we observe a significant drop in performances with auto-calibration. Simultaneously, CADet performs well on detecting unknown classes, although not beating the best existing methods on iNaturalist.

Notably, applying CADet to a supervised network achieves state-of-the-art performances on iNaturalist with ResNet50 architecture, suggesting CADet can be a reasonable standalone detection method on some benchmarks, independently of contrastive learning. In addition, the poor performances of the supervised network on ImageNet-O and adversarial attacks show that contrastive learning is essential to address the trade-off between different type of anomalies.

Overall, our results show CADet achieves an excellent trade-off when considering both adversarial and label-based OOD samples.

## 6.2 THE PREDICTIVE POWER OF IN-SIMILARITIES AND OUT-SIMILARITIES

Table 5 reports the mean and variance of $m^{in}$ and $m^{out}$, and the rescaled mean $\gamma m^{out}$ across all distributions. Interestingly, we see that out-similarities $m^{out}$ better discriminate label-based OOD samples, while in-similarities $m^{in}$ better discriminate adversarial perturbations. Combining in-similarities and out-similarities is thus an essential component to simultaneously detect adversarial perturbations and unknown classes.

## 6.3 LIMITATIONS

**Computational cost:** To perform detection with CADet, we need to compute the features for a certain number of transformations of the test sample, incurring significant overhead. Figure 2 shows that reducing the number of transformations to minimize computational cost may not significantly affect performances. While the calibration step can be expensive, we note that it only needs to be run once for a given in-distribution. The coefficient $\gamma$ and scores are all one-dimensional values that can be easily stored, and we purposely use a small number of validation samples $|\{X_{val}^{(1)}\}| = 300$ to make their embedding easy to memorize.

Table 5: Mean and variance of $m^{in}$ and $m^{out}$.

|      |              | IN-1K   | iNat    | IN-O    | PGD     | CW      | FGSM    |
|------|--------------|---------|---------|---------|---------|---------|---------|
|      | $m^{in}$     | 0.972   | 0.967   | 0.969   | 0.954   | 0.954   | 0.948   |
| Mean | $m^{out}$    | 0.321   | 0.296   | 0.275   | 0.306   | 0.302   | 0.311   |
|      | $\gamma m^{out}$ | 0.071 | 0.066   | 0.061   | 0.068   | 0.067   | 0.069   |
| Var  | $m^{in}$     | 8.3e-05 | 7.8e-05 | 1.0e-04 | 2.1e-04 | 2.0e-04 | 2.1e-04 |
|      | $m^{out}$    | 1.7e-03 | 7.0e-04 | 2.3e-03 | 7.0e-04 | 1.7e-03 | 1.1e-03 |

**Architecture scale:** as self-supervised contrastive learning is computationally expensive, we only evaluated our method on a ResNet50 architecture. In Wang et al. (2022), the authors achieve significantly superior performances when using larger, recent architectures. The performances achieved with a ResNet50 are insufficient for real-world usage, and the question of how our method would scale to larger architectures remains open.

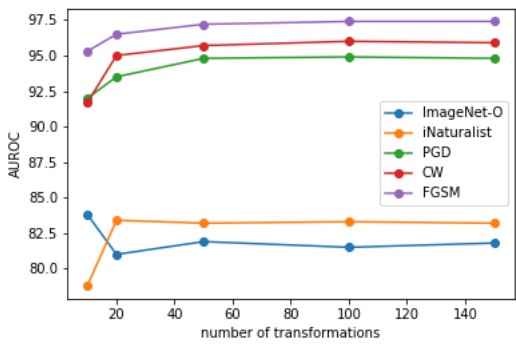

Figure 2: AUROC score of CADet against the number of transformations.

## 6.4 FUTURE DIRECTIONS

While spurious correlations with background features are a problem in supervised learning, it is aggravated in self-supervised contrastive learning, where background features are highly relevant to the training task. We conjecture the poor performances of CADet on iNaturalist OOD detection are explained by the background similarities with ImageNet images, obfuscating the differences in relevant features. A natural way to alleviate this issue is to incorporate background transformations to the training pipeline, as was successfully applied in Ma et al. (2018). This process would come at the cost of being unable to detect shifts in background distributions, but such a case is generally less relevant to deployed systems. We leave to future work the exploration of how background transformations could affect the capabilities of CADet.

## 6.5 CONCLUSION

We have presented CADet, a method for both OOD and adversarial detection based on self-supervised contrastive learning. CADet achieves an excellent trade-off in detection power across different anomaly types. Additionally, we discussed how MMD could be leveraged with contrastive learning to assess distributional discrepancies between two sets of samples.

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
