# OpenReview forum: "CADet: Fully Self-Supervised Anomaly Detection With Contrastive Learning"
_ICLR.cc/2023/Conference — Submitted to ICLR 2023_

### Official Review · Reviewer_TDRY · 2022-10-22

**Confidence:** 4
**Correctness:** 2
**Technical Novelty And Significance:** 3
**Empirical Novelty And Significance:** 3
**Recommendation:** 6

**Clarity, Quality, Novelty And Reproducibility:**

It is well written and easily readable.

it has two small improvements over standard MMD, which seem to be novel.

self-supervision for outlier detection as a whole is not novel  (https://openaccess.thecvf.com/content/CVPR2021/papers/Li_CutPaste_Self-Supervised_Learning_for_Anomaly_Detection_and_Localization_CVPR_2021_paper.pdf , https://arxiv.org/abs/2103.12051 ) but its evaluation in the context of MMD seems to have novelty, which is incremental..

**Strength And Weaknesses:**

Strengths:
1 well written paper
2 experiments with out-of sample and adversarial data
3 an improvement in detection in setups where one has a set of test samples due to the two statistical changes
4 incremental novelty in combining self-supervision and MMD

weaknesses:

1 There might be problems in the formulation of the third contribution:

1.1 First of all, a naming criticism: The similarity is fixed. This the authors also acknowledge themself. This is okay but it is unclear why it is named contrastive, as no change or training of similarity is performed based on the augmentations of the test sample. Yes, they initialize the similarity using contrastive learning, but the test sample is not involved in this first step. It is averaging over copies obtained by data augmentation, and thus not more contrastive than the original MMD with self-supervised initialization.

Contrastively-initialized outlier detection is more precise here.

1.2 Secondly, and most importantly, equation (6) departs from the idea of an MMD score substantially. The MMD score as in eq (4) measures a difference between self-similarities and cross-similarities. Equation (6) performs an addition where MMD as in eq (4) would perform a subtraction.

Comparing to eq (4) also a self-similarity term between validation data is missing. For thresholding it can be omitted, but it would impact the variance of the estimators. However the main issue is the mentionned addition in place of substraction of the cross-similarity.

equation (6) becomes by that a statistic without any motivation behind it. Why it is chosen as addition when (4) suggest a substraction ?

1.3 Furthermore by matching equation (7) with algorithm 2 (on page 7) one can see that the calibration to compute gamma uses only data from the same in-distribution. But then asymptotically gamma should become 1, as m^{out} and m^{in} are computed over data drawn from the same distribution an thus their variance should be the same.

If gamma is not 1, as in Table 5, then something in the implementation seems to be unexpected (although gamma seems to be a constant across the datasets in Table 5).

1.2 and 1.3 look as if they wanted to do something slightly different. gamma becomes non-trivial if it involves a mix out validation and augmented variants of the test sample.


2 it would be good to try adversarial detection with other types of adversarial attacks, which are generated by different principles

3 self-supervised outlier detection is not novel itself (https://openaccess.thecvf.com/content/CVPR2021/papers/Li_CutPaste_Self-Supervised_Learning_for_Anomaly_Detection_and_Localization_CVPR_2021_paper.pdf , https://arxiv.org/abs/2103.12051 ), but see strengths, point 4

**Summary Of The Paper:**

The authors present research on the intersection of outlier detection using MMD on two sets of samples and self-supervied pretraining.
First of all they use self-supervision to pretrain a similarity function which is then to be used for outlier detection using MMD and derived statistics.

They present two improvements over Liu et al. 'Learning deep kernels for non-parametric two-sample tests.' when computing statistics. One is a better calibration of the p-value calculation. The other is a different way to compute the MMD statistic, in which they double the size of the validation set, and likely are able to achieve a lower variance of the p-scores if the test distribution is different from the validation distribution.

They show two results in these parts. One is stronger evidence for a distribution shift between CIFAR10 and CIFAR10.1 than shown in previous work. The second are results for outlier detection against sets of outliers taken form other datasets and obtained by adversarial optimization.

The third contribution they state as contrastive anomaly detection. This is meant to identify a single test sample whether it could be an outlier.

In this they take the test sample, and create a set of augmented variants from it, using augmentations similar to those used in self supervised pretraining. They do the same with a set of validation samples. Then they define a statistic somewhat reminiscent of the one used in MMD to obtain a score, which then is used to compute a p-value. The p-value is based on comparing scores computed between two augmented validation sets from the in distribution, and a score computed between one augmented validation set and one set obtained by augmenting the test sample.

They show results in Table 3 and 4 which are mixed for outlier detection and good for adversarial detection.


**Summary Of The Review:**

The paper has two smaller statistical improvements over MMD. It advocates the use of self-supervision for outlier detection when combined with MMD. That is an insight.

The strongest issue with respect to acceptance in the eyes of this reviewer is the last contribution on single sample outlier detection which seems to depart from MMD without any motivation for doing so.

*edit* after reading the rebuttal, the reviewer increased his score from 5 to 6 ,

however, if accepted to ICLR please clarify that it is inspired by MMD but differs in that it gives up / departs from the geometric interpretation used in MMD (when switching the sign) + provide the motivation stated in the rebuttal.

---

> ### Author Response · Authors · 2022-11-10
> **authors' answer**
>
> We thank the reviewers for their careful consideration and valuable feedback. We are glad they all found our work well-written.
>
> Several reviewers have made suggestions and asked questions leading to discussions in the rebuttals that are not included in our submission, and will be relevant to the reader. Therefore, we propose to follow the suggestion of reviewer qf3u and move the pseudocode of algorithms 2 and 3 to the appendix, making space to include discussions suggested by the reviewers, and outlined in our answers.
>
> > weaknesses
>
> > 1.1
>
> We agree with the referee and propose to improve our terminology in the suggested way. While contrastive learning is only used for the training of the backbone, our idea is that the similarity function induced by such models (by taking the cosine of their learned representations) has specific properties that we propose to leverage in this work. We believe our methods to be specifically designed for backbones trained under contrastive scheme, and that is the reason we named it ‘anomaly detection with contrastive learning’. However, since the detection method itself is not contrastive, we will follow the referee’s suggestion.
>
> > 1.2
>
> Regarding the self-similarity term between validation data, we omitted it because it is a constant, and therefore does not affect the outcome. It is unclear to us why it would impact the variance of estimators. Note that the set $X_{val}^{(1)}$ and the transformation of its samples are fixed, because recomputing the features on each transformation of each sample at every inference step would be computationally unreasonable.
>
> Regarding the reason the addition becomes a subtraction, it is because while CADet is directly inspired by MMD through the quantities involved, it requires significant changes to perform detection using a single sample. While augmenting the test sample with transformations is a workaround, it will result in a different behavior. In MMD, assuming the same distribution, the expectation of cross-similarity and intra-similarity is the same. This is not the case in our setting, even on in-distribution samples, because cross-similarity will be averaged across samples and transformations, while intra-similarity is only averaged across transformations of the same sample. Thus we can not simply take the difference between the two.
>
> CADet therefore uses these quantities in a different manner: the main intuition is that self-supervised contrastive learning specifically trains models to maximize the cosine similarity of representations for transformations of an in-distribution sample. We can thus expect that on out-of-distribution samples, the model will not perform as well in that regard, therefore leading to smaller $m_{out}$. Similarly, outliers can be expected to have lower similarity to genuine in-distribution samples, leading to smaller expectation of $m_{in}$. Since both scores are expected to be smaller on outliers, we take their addition. Both quantities are inspired by MMD, but we leverage them in an entirely novel way.
>
> This intuition is missing from our work, which is a mistake seriously impeding its interpretability. We propose to discuss it at the beginning of Section 5.
>
> > 1.3
>
> This point is related to the above discussion: $m_{in}$ is averaged over different transformations AND samples, while $m_{out}$ is only averaged over different transformations of a single sample. Therefore, they will have different variances and means even asymptotically and on in-distribution samples. For instance, gamma will be affected by the variance of learned representations with respect to the sampling of in-distribution data. This justifies the need to auto-calibrate gamma.
>
> > 2
>
> We agree with the referee, and while we believe our evaluations are meaningful, they would be better supported if we had included e.g. gradient-free attacks. However, an exhaustive empirical analysis is not feasible and there will always be some attacks left out.
>
> > 3
>
> We agree with the referee, and would like to point out that Section 2 (Related Work) includes a “Contrastive learning for OOD detection” paragraph where we discuss many such works, including the second work mentioned by the referee. The first one is missing from our bibliography and is relevant, so we will cite it in this paragraph. One major difference in our work is that to the best of our knowledge, existing self-supervised outlier detection works focus on detecting novel classes. They also tend to address easier detection problems, as opposed to the fine-grained novel classes detection benchmarks used in our work. Compared to other self-supervised detection methods, we perform detection of a wider and more difficult scope of outliers, getting one step closer to real-world applicability.

---

> > ### Comment · Reviewer_TDRY · 2022-12-04
> > **1.3**
> >
> > the authors write that:
> >
> > "This point is related to the above discussion: $m_{in}$ is averaged over different transformations AND samples, while is $m_{out}$ only averaged over different transformations of a single sample. Therefore, they will have different variances and means even asymptotically and on in-distribution samples. For instance, gamma will be affected by the variance of learned representations with respect to the sampling of in-distribution data. This justifies the need to auto-calibrate gamma."
> >
> > but $m_{in}$ and $m_{out}$ in eq 5 seems to say the opposite ?
> >
> > "we compute ntrs random transformations xtest
> > i of xtest"
> >
> > $m_{in}$ in eq 5 averages over only j and i, thus transformations of $x^{test}$
> > whereas in eq 6 the average is over samples and transformations, as an average over $X_{val}$ is computed ?
> >
> > so $m_{in}$ averages only over transformations of  one sample, and $m_{out}$ over transformations and samples ?
> >
> > In that case $\gamma$ can be $\gamma \neq 1$ indeed.

---

> > > ### Author Response · Authors · 2022-12-06
> > > **re: 1.3**
> > >
> > > We indeed made a mistake in our answer: as the reviewer point out, it is $m_{in}$ that is averaged only on different transformations and $m_{out}$ that is averaged on both samples and transformations. We interchanged $m_{in}$ and $m_{out}$, but the underlying argument remains the same : they should not be expected to have the same variance even in-distribution (thus $\gamma\neq 1$) because $m_{out}$ is measured between different samples, and thus is expected to have smaller mean and larger variance than $m_{in}$, which is experimentally confirmed in Table 5

---

> > > > ### Comment · Reviewer_TDRY · 2022-12-06
> > > > **update**
> > > >
> > > >
> > > > after reading the rebuttal, the reviewer increased the score from 5 to 6 ,
> > > >
> > > > however, if accepted to ICLR, please clarify (1) that it is inspired by MMD but differs in that it gives up / departs from the geometric interpretation used in MMD (when switching the sign) + (2) provide the motivation stated in the rebuttal.
> > > >
> > > > The reviewer does not see the departure from the original MMD motivation as a sufficient reason to reject this paper.

---

> > ### Comment · Reviewer_TDRY · 2022-12-04
> > **1.2**
> >
> > By looking at eq 5:
> >
> > We can thus expect that on out-of-distribution samples, the model will not perform as well in that regard, therefore leading to smaller $m_{out}$.
> >
> > --> ok, makes sense, if $X_{val}$ are inliers, and $x^{test}$ an outlier, then $m_{out}$ will be small, as the cross similarity terms will be small.
> >
> > Similarly, outliers can be expected to have lower similarity to genuine in-distribution samples, leading to smaller expectation of $m_{in}$.
> >
> > --> $m_{in}$ measures a similarity between transformations of a single sample itself, thus it can be high even on reasonable outliers. If it would be lower, it should be proven experimentally.
> >
> > if you sum these terms, you do not compare within distribution similarities versus cross-sample similarities anymore, as you sum them. So it departs from the geometric idea of MMD (a kernel-induced distance between two sets of samples).
> > It is then MMD-inspired, yes, but not MMD-based.

---

> > > ### Author Response · Authors · 2022-12-06
> > > **re: 1.2**
> > >
> > > Thank you for taking the time to read our answer !
> > >
> > > - Regarding the first point, indeed m_out is expected to be smaller on outliers due to lower cross similarity with inliers from X_val. However in the case of m_in, it is not due to lower similarity to genuine in-distribution samples, since m_in does not consider validation samples (m_in stands for intra-sample similarity, not in-distribution, we had not realized that this terminology might be confusing).
> > > m_out can be seen as estimating expectation of similarity between a random transformation of x_test and a random transformation of an inlier, and m_in as estimating expectation of similarity between two random transformations of x_test. The reason why m_in is expected to be lower is detailed in the following point.
> > >
> > > - m_in could be high on outliers, but it is unlikely, especially when using a network trained with contrastive learning. The main idea here is that a network will generally perform better (in term of training objective) in-distribution than out-of-distribution. Since a part of the self-supervised contrastive loss we use to train the network is to maximize the similarity between transformations of a same sample, our contrastive network is trained to maximize m_in. Therefore, we expect m_in to be higher in-distribution than out-of-distribution. This is a major intuition behind the use of self-supervised contrastive learning to train the backbone.
> > > These results are already confirmed experimentally in Table 5, where we observe that m_in has a higher mean in-distribution than for any distributional shift we consider. The difference is particularly large (0.02) on adversarial attacks, which is a significant gap considering the small variance.
> > >
> > > It is absolutely correct to say CADet is inspired by MMD, since both quantities involved are related to the quantities computed by MMD, but is significantly different as these quantities are leveraged in a novel setting and in a novel manner. Section 4 shows that MMD two-sample test with a learned kernel has a strong testing power against these distributional shifts, and CADet is a method based on the MMD quantities to perform single-sample detection -- but while it is based on the quantities involved in MMD, it is indeed significantly different from MMD itself and its underlying theory

---

### Official Review · Reviewer_FoD6 · 2022-10-25

**Confidence:** 4
**Correctness:** 3
**Technical Novelty And Significance:** 4
**Empirical Novelty And Significance:** 3
**Recommendation:** 6

**Clarity, Quality, Novelty And Reproducibility:**

The paper is overall very clear, and the proposed method is presented in a sound way. I believe the successful application of MMD-based two-sample testing on OOD detection has strong novelty and significance. The paper contains enough information to reproduce the empirical results. Meanwhile, I do have a few concerns regarding the empirical results, particularly in Table 4.

**Strength And Weaknesses:**

# Strength

* As far as I know, the application of MMD-based two-sample testing on OOD detection is a novel approach. There have been some efforts to apply the two-sample testing idea to OOD detection, but they were not as successful as this paper.
* The paper is well-written. The expositions are clear and easy to follow.
* The paper does a good job of summarizing the existing literature and providing the technical information required to reproduce the experiments.
* The paper provides a helpful discussion on the limitations and future directions of the work in Section 6.3.

# Weaknesses

I think the followings are more like questions rather than weaknesses.

* ("Supervised" adversarial attack) Although it is stated in Section 5.1, it would be nicer to emphasize that the adversarial samples considered in this paper are generated with respect to a supervised classifier. The reason is that an adversarial sample can also be generated with respect to an OOD detector, and OOD detectors are known to be very vulnerable to such attacks, see for example, https://arxiv.org/abs/2106.04260 . Nonetheless, limiting the scope of the paper to the "supervised" adversarial attack would not be the reason for rejection, I think.
* (Adversarial detection performance of contrastive representation) I am worried that the comparison between supervised representation and contrastive representation in Table 4 is not fair, because these adversarial samples are generated with respect to the supervised network being tested. From the information provided in the paper, it is not clear whether the attack would transfer to a network trained with contrastive learning. Even though it translates well, the attack is usually the strongest against a network the attack is targeting. Therefore, a supervised network can be said to be handicapped. I believe it is a little bit too early to conclude that contrastive representation is more effective in adversarial detection. A few more supporting arguments should help resolve this issue.
* (The number of inlier samples in testing) For me, it is surprising that so few inlier samples are needed to perform successful two-sample testing with respect to outlier classes, and I wonder if there is a good explanation regarding this. Naively, if there are 1,000 classes in the in-distribution dataset, shouldn't we need at least 1,000 samples that represent each class?

**Summary Of The Paper:**

This paper presents a novel out-of-distribution detection algorithm inspired by MMD-based two-sample testing. The paper first shows that contrastive learning, when used with MMD two-sample test, can be highly effective in recognizing distribution shifts in image statistics. Then, the paper extends this finding to build an out-of-distribution detection algorithm that can detect images that contain an object that is not in the training set or is adversarially modified to fool an image classifier.

**Summary Of The Review:**

It is a well-written paper with algorithmically novel contributions. It would be a more appealing paper if the performance in Table 3 was stronger and the fair comparisons were made in Table 4.

---

> ### Author Response · Authors · 2022-11-10
> **Authors' answer 1/2**
>
> We thank the reviewers for their careful consideration and valuable feedback. We are glad they all found our work well-written.
>
> Several reviewers have made suggestions and asked questions leading to discussions in the rebuttals that are not included in our submission, and will be relevant to the reader. Therefore, we propose to follow the suggestion of reviewer qf3u and move the pseudocode of algorithms 2 and 3 to the appendix, making space to include discussions suggested by the reviewers, and outlined in our answers.
>
> > ("Supervised" adversarial attack)
>
> We will insist on that point and clarify as much as possible. It seems indeed likely that adversarial attacks could be generated to maximize our detection score and thus prevent detection with high efficiency. We will cite and comment on the paper mentioned by the referee, as we were not aware of it and it is highly relevant to our work. Several other prior works have explored robustness to attacks on the detector and shown this vulnerability, such as https://arxiv.org/pdf/1705.07263.pdf. One major issue with the study of such type of robustness is that attacks targeted at the detector need to be designed specifically for each detection method, and the apparent robustness of a detector to a targeted attack might very well be due to poor design of the attack. It is very difficult to compare the robustness of different detectors to such attacks, or even claim the robustness of a single detector, because there is no objective evaluation of the ‘design quality’ of chosen targeted attacks. For these reasons, we believe our setting is the most reasonable to allow comparisons. Moreover, while many prior works have shown that adversarial attacks on images often transfer rather well across models, increasing vulnerability even in black-box settings, it is reasonable to expect these targeted attacks wouldn’t transfer across detection methods, due to the variety of their approaches – though this intuition is not substantiated by rigorous experiments. Finally, having to fool both the detector and the classifier is typically a harder task than fooling only the classifier, due to adding a novel objective. Attacks that fool the detector but not the classifier cannot reasonably be considered as successful attacks. Thus, while still vulnerable to attacks targeted at the detector, including an OOD detection method will typically make the whole system more difficult to attack.
>
> > (Adversarial detection performance of contrastive representation)
>
> The literature seems to be split on this question and we believe further studies are required to conclude on whether this setting makes detection harder or not. For instance https://openaccess.thecvf.com/content/ICCV2021/papers/Abusnaina_Adversarial_Example_Detection_Using_Latent_Neighborhood_Graph_ICCV_2021_paper.pdf claims that such attacks are actually harder to detect because they generate smaller perturbations of the features. We believe this argument is weak, because while features may suffer from smaller perturbations, their directions may not match the clusters corresponding to other classes, and thus might be easier to categorize as outliers. While we do not explicitly present such discussion in our work, Table 4 provides some insights on this topic: Odin achieves worse performances using a supervised backbone, while Hu et al. achieves better performances. Our method also achieves substantially better performances on a contrastive backbone, but this may not be related to detection difficulty, since our approach is specifically designed for models trained in a contrastive setting. Overall, we agree direct comparisons between performances with a supervised backbone and with a contrastive backbone are unfair. This is why in Table 4, we present performances on both supervised and contrastive backbones for each method. We believe this makes the comparison fair by comparing the performances achieved by different methods on the same backbone. In particular, CADet very significantly outperforms baselines even when evaluating on the same contrastive backbone, which we believe to make our results more fair and convincing. Another supporting point is that despite CADet being designed for contrastive backbones, its performance using a supervised backbone is not bad. Indeed, while Hu et al. achieves higher AUROC on a supervised backbone, it also uses adversarial examples and extensive hyperparameter search to tune their metric, which is bound to overfit this specific type of distributional shift. When tuning Hu et al.’s metric only on in-distribution data with the same method we use (balancing variance of all terms), CADet significantly outperforms both baselines even on supervised backbones.

---

> > ### Author Response · Authors · 2022-11-10
> > **authors' answer 2/2**
> >
> > > (The number of inlier samples in testing)
> >
> > This is an excellent point. In practice, we observed very little variance with respect to the sampling of $X_{val}^{(1)}$, and concluded that the 300 random samples are sufficient to capture the properties of in-distribution classes, due to overlapping features in ImageNet classes. However, increasing the size of $X_{val}^{(1)}$ may indeed lead to less false positives related to in-distribution classes with little similarity to the classes present in $X_{val}^{(1)}$. However, since memory and computational cost is probably the main practical limitation of CADet, as discussed in the limitations section, we believe increasing the size of $X_{val}^{(1)}$ is an unfavorable tradeoff. The computation of the features only needs to be performed once, but a matrix of shape $(|X_{val}^{(1)}| n_{trs}, d_{emb})$ (where $d_{emb}$ is the embedding dimension) needs to be stored in memory and multiplied to another large matrix at each inference, therefore limiting the size of $X_{val}^{(1)}$ is very beneficial in terms of complexity.

---

### Official Review · Reviewer_qf3u · 2022-10-31

**Confidence:** 4
**Correctness:** 4
**Technical Novelty And Significance:** 2
**Empirical Novelty And Significance:** 3
**Recommendation:** 5

**Clarity, Quality, Novelty And Reproducibility:**

+ Overall, the paper is well written and easy-to-follow.
+ All training details seem to be given for the reproducibility of results.

- I find the technical and methodological novelty of the paper to be overall rather low, presenting a combination of existing self-supervised embedding learning (SimCLRv2) with a MMD two-sample test. Novelty is given in extending this idea to single samples via augmentation and the proposed extension of MMD.
- The space used in the main paper could be optimized in my opinion. Some background could maybe trimmed a bit to provide more space/details on the proposed methods (most content on pages 1--4 is background). Algorithms 2 + 3 could be put into the appendix.

---

*Additional Comments*
* p.1: "Anomaly detection methods generally rely on [...]" I'd rather say OOD detection methods here, since anomaly detection is more general, and OOD detection is a specific anomaly detection problem for which specific methods have been proposed.
* p.2: "Contributions: Our main contributions are as follows:" Double colon (remove first).

**Strength And Weaknesses:**

+ I find it interesting to see that self-supervised embeddings improve adversarial attack detection over using supervised embeddings, also for other previous methods. I'm not aware of previous work making this observation.
+ The paper is well integrated into the existing literature, making a comparison to recently established OOD methods.

- Overall, I find the experimental results to be not too convincing. For detecting novel classes, the proposed method does not show a consistent improvement over previous methods (the supervised embeddings also being better over the self-supervised one in 2/3 cases). The adversarial examples detection is more convincing, which I think should be extended/elaborated on.
- While CADet empirically seems to work particularly well for adversarial attacks, I am missing some explanation or intuition on why this might be the case. I think the paper might be stronger focusing and digging deeper on this case, since the differences over previous methods here are intriguing.

**Summary Of The Paper:**

The paper studies the detection of out-of-distribution (OOD) instances (novel classes and adversarial attacks) in vision using similarities learned via self-supervised contrastive learning (SimCLRv2) with a maximum mean discrepancy (MMD) two-sample test. This approach is first used to show that there exists a significant distributional shift between CIFAR10 and CIFAR10.1 (which has been proposed as a corrected update of CIFAR10). Then, an extension of the approach for single samples is proposed via using augmentations of a sample to create a set, called CADet (Contrastive Anomaly Detection). An experimental evaluation on OOD detection of (i) novel classes (ImageNet vs. iNaturalist, ImageNet-O) and (ii) adversarial examples (generated with PGD, CW, and FGSM attacks) shows that CADet performs similar on detecting unknown classes and favorably on detecting adversarial attacks over previous methods.

**Summary Of The Review:**

The paper includes interesting findings regarding differences between using self-supervised vs. supervised embeddings for OOD, in particular wrt adversarial examples, but is limited in providing further analysis and insights into why one might expect the shown improvements in detecting adversarial examples.

---

> ### Author Response · Authors · 2022-11-10
> **Authors' answer 1/2**
>
> We thank the reviewers for their careful consideration and valuable feedback. We are glad they all found our work well-written.
>
> Several reviewers have made suggestions and asked questions leading to discussions in the rebuttals that are not included in our submission, and will be relevant to the reader. Therefore, we propose to follow the reviewer's suggestion and move the pseudocode of algorithms 2 and 3 to the appendix, making space to include discussions suggested by the reviewers, and outlined in our answers.
>
> > Overall, I find the experimental results to be not too convincing. For detecting novel classes, the proposed method does not show a consistent improvement over previous methods (the supervised embeddings also being better over the self-supervised one in 2/3 cases). The adversarial examples detection is more convincing, which I think should be extended/elaborated on.
>
> While our method does not outperform existing works on novel classes (and as the reviewer noted, even with our method, a supervised backbone performs slightly better on average), the strength of our proposed approach is the simultaneous detection of adversarial attacks and novel classes. We compare our performances on novel classes detection to existing works specialized on that task, and compare our performances on adversarial detection to existing works specialized on that task. Being able to compare on novel classes while simultaneously outperforming on adversarial detection, without any fine-tuning or adaptation of the method is in our eyes a significant improvement, making our method one step closer to real world viability, where the type of distributional shift is typically not known in advance. In other words, the main strength of CADet is in the increase of the scope of detection rather than purely the increase of specialized performances.
>
> Another point on which we may have not sufficiently insisted is that our calibration is made purely on in-distribution samples. Most existing methods calibrate directly on OOD samples, which is a significantly easier setting, but unrealistic in practice. Some methods calibrate on OOD samples of different test datasets than the ones used for evaluation, but with the same type of distributional shift (e.g. novel classes). The only work we are aware of that calibrates their scores in a similarly difficult setting is “ViM: Out-of-Distribution With Virtual Logits Matching”, which is a significant improvement but is designed specifically for novel classes detection. For instance in Table 4, the method of Hu and al. can be calibrated in the same way as ours, which results in a significant drop of performances and illustrates the difficulty of our setting.
>
> > While CADet empirically seems to work particularly well for adversarial attacks, I am missing some explanation or intuition on why this might be the case. I think the paper might be stronger focusing and digging deeper on this case, since the differences over previous methods here are intriguing.
>
> Our intuition on the reason CADet + contrastive self-supervision works so well on adversarial detection is that contrastive models have been trained to compute similar embeddings on random crops of a given image. Adversarial attacks will perturb the features in a way that changes the classification on the full image, but this will also affect the capability of the network to match random crops of the image, greatly reducing $m_{in}$. This intuition is supported by the fact that adversarial detection improves if we were to rely only on $m_{in}$ (or by manually setting gamma to a small value). We do not report these results as they would be ‘overfitting’ adversarial shifts, which goes against our goal of a general purpose method.
>
> Similarly, having access to labels helps supervised models learn label-specific expressive features, which are useful to distinguish novel classes that are similar to in-distribution classes, as is the case with fine-grained detection such as the iNaturalist benchmark. This explains why self-supervised networks perform comparatively worse on novel classes benchmark. It seems clear to us, however, that they achieve the best tradeoff across detection tasks.
>
> This explanation is only an hypothesis, but we propose to follow the reviewer’s recommendation and include it in our discussion (while emphasizing its speculative nature).

---

> > ### Author Response · Authors · 2022-11-10
> > **authors' answer 2/2**
> >
> > > I find the technical and methodological novelty of the paper to be overall rather low, presenting a combination of existing self-supervised embedding learning (SimCLRv2) with a MMD two-sample test. Novelty is given in extending this idea to single samples via augmentation and the proposed extension of MMD.
> >
> > Regarding the novelty, we would like to emphasize that we are, to the best of our knowledge, the first work to explore MMD two-sample tests with a pretrained kernel, and we achieve significantly better results than existing works on the CIFAR10 vs CIFAR10.1 task using this approach. We also propose a small improvement to the method (MMD-CC).
> >
> > The adaptation to single sample detection with CADet is also entirely new, as the reviewer notes. It is also, to the best of our knowledge, the first work exploring contrastive learning for adversarial detection.
> >
> > > The space used in the main paper could be optimized in my opinion. Some background could maybe trimmed a bit to provide more space/details on the proposed methods (most content on pages 1--4 is background). Algorithms 2 + 3 could be put into the appendix.
> >
> > We agree with the reviewer and we propose to move Algorithms 2 and 3 to the appendix to incorporate the reviewers' suggestions, including the discussion on the reason CADet performs so well on adversarial detection.
> >
> > > Additional Comments
> >
> > > p.1: "Anomaly detection methods generally rely on [...]" I'd rather say OOD detection methods here, since anomaly detection is more general, and OOD detection is a specific anomaly detection problem for which specific methods have been proposed.
> >
> > We agree with the reviewer and will make that change. Terminology is a bit difficult in our setting as many researchers do not consider adversarial attacks as a type of distributional shift. However, since we have clarified in our introduction that we treat adversarial perturbations as a type of OOD samples, our concerns are probably unjustified.
> >
> > > p.2: "Contributions: Our main contributions are as follows:" Double colon (remove first).
> >
> > We will make that change.

---

### Official Review · Reviewer_txxW · 2022-11-03

**Confidence:** 2
**Correctness:** 3
**Technical Novelty And Significance:** 3
**Empirical Novelty And Significance:** 3
**Recommendation:** 6

**Clarity, Quality, Novelty And Reproducibility:**

### Clarity

1 Are S_p(1) and S_p(2) same size in Algorithm 1? Are they half size of S_q?

2 Section 4.1 mentions that 100 samplings are used for the experiment. Does it mean 50/50 from in/out of distributions (total 100 test cases) or 100/100 (200 test cases)? Or is the ratio of in/out distribution cases not equal to 0.5? (not equal number of test cases for in and out-of-distributions?)

3 What is the ‘learned similarity’? The similarity is defined as cosine similarity in Equation (1). Does learned similarity denote the cosine similarity with learned feature representation?

### Quality

The paper is well-written although there are some missing ablation studies and experiment details.

### Novelty

The paper’s idea mainly relies on MMD and is technically similar to classic MMD for out-of-distribution detection.
However, the paper proposes several parts to adapt the idea to anomaly detection.

### Reproducibility

Although most of the necessary processes are described in the paper, it will not be easy to reproduce the paper without making the code publicly available since the algorithm requires multiple steps to compute and multiple hyper-parameters (including sample set sizes) are involved in reproducing the experiments.


**Strength And Weaknesses:**

## Strength

### Various experiments support the effectiveness of the method

### Diverse applications

Out-of-distribution detection from two sample sets, single instance anomaly detection, and adversarial attach detections are shown in the experiment section.

### Diverse datasets are used for the experiments.

Four datasets are used for out-of-distribution detection tasks (tasks that assume a set of test samples are given).  Two image datasets are used for anomaly detection.
Three adversarial scenarios are shown (on the ImageNet validation set) for anomaly detection tasks (single outlier instance detection).

### Ablation studies

Some necessary ablation studies are presented in the paper including varying sample set sizes (Figure1 and Table1), and the number of transformations used for anomaly detection (figure 2).

### The limitations of the paper are addressed.



## Weakness

### Computational cost

The paper also addresses their computational cost requirement. Since two sample test is a statistical test on two sample sets, their computation requires quadratic order in terms of sample sizes. Moreover, the paper proposes to repeat this test multiple times to reduce the MMD variance.

### Pretraining on a large-scale dataset required

The method relies on well trained feature representation. The experiments are performed on a pre-trained ResNet 50 or a self-trained ResNet 50.

### Missing ablation studies

1 Computing calibration parameter gamma in Eq (7) costs n_trs * n_trs * |X_val1| * ||X_val2| computations. How much is this calibration process important? What is the performance gain from this process? What if gamma=1 is used without any calibration process?

2 The paper uses hypothesis testing instead of directly using the score in equation (8). How much performance gain was obtained from this?

### (Minor point) The method may be sensitive to different transformations.

In the first paragraph of section 5.1, transformations except ‘color jittering, gaussian blur, and gray scaling’ are used. What is the rationale behind this? Is the method sensitive to different types of transformations?


**Summary Of The Paper:**

The paper proposes two sample test method that measures the maximum mean discrepancy (MMD) of two sample sets for out-of-distribution sample set detection. The main idea is to use self-supervised features for MMD. On top of the self-supervised features, the paper proposes a method called MMD-CC (MMD with clean calibration) that reduces the variance of MMD in the cost of computations which is effective when sample set sizes are small.
Then, the authors propose a method to apply MMD and MMD-CC to a single outlier instance detection (anomaly detection) by simulating a two sample test on a single test sample by populating the test sample with multiple transformations (augmentations).
Two terms are used to measure anomaly score; intra similarity score which measures the similarity between transformed test samples, and out similarity score which measures the similarity between transformed training and test samples. The calibration step is proposed to find a balancing hyper-parameter between the two similarity score terms. Instead of using the similarity score directly as an anomaly score, the paper proposes to perform hypothesis testing and use the p-value as an anomaly score. The proposed methods are evaluated on 4 datasets for out-of-distribution detection tasks (a task that assumes a set of test samples are given), and evaluated on two image datasets and 3 adversarial scenarios (on the ImageNet validation set) for anomaly detection tasks (single instance detection).


**Summary Of The Review:**

The paper proposes a two-sample-test-based out-of-distribution detection and anomaly detection.
The experiments are extensive with meaningful results. Although there are a few missing ablation studies and details, the paper is well-written overall.

---

> ### Author Response · Authors · 2022-11-09
> **Author's answer 1/2**
>
> We thank the reviewers for their careful consideration and valuable feedback. We are glad they all found our work well-written.
>
> Several reviewers have made suggestions and asked questions leading to discussions in the rebuttals that are not included in our submission, and will be relevant to the reader. Therefore, we propose to follow the suggestion of reviewer qf3u and move the pseudocode of algorithms 2 and 3 to the appendix, making space to include discussions suggested by the reviewers, and outlined in our answers.
>
> > ### Computational cost
>
> It is correct that to perform MMD two sample tests, we need to perform a number of inner products quadratic in the sample size, and need to repeat that process to get reasonable variance. However, these scalar products are only performed on the embeddings computed by the neural network, which are typically of reasonable dimension ($d_{emb}\approx 10^3$).
>
> Given the embeddings, computing the MMD score is equivalent to a matrix product $A^{T}A$ where A is a matrix of shape $(d_{emb}, 2n_{samples})$. In practice, this dot product is very efficient and takes a negligible amount of time compared to computing the embeddings, which requires a single forward pass of the neural network on each sample.
>
> For instance, performing MMD-CC with a ResNet-50 on Cifar-10.1 vs Cifar-10 with $n=2000$ samples and $p=500$ permutations (Table 1) with a given sampling requires less than a minute with a single A100 GPU (for both embeddings computations and MMD score computations). Of course, this computation time could become unreasonable as n grows extremely large, but Table 1 seems to indicate that reaching that sort of scale is unnecessary.
>
> While section 6.3 addresses the computational limitations of CADet, it does not discuss computation cost of MMD-CC. We propose to add a paragraph at the end of section 4 to explain that the computational complexity is in $O(n T_{fp} + n^2 p d_{emb} )$, where $T_{fp}$ is the complexity of a single forward pass,  and that both terms are typically cheap on modern hardware in reasonable settings.
>
> > ### Pretraining on a large-scale dataset required
>
> That is correct. Note that any model pre-trained on the in-distribution data can be used, no finetuning is required on out-of-distribution data. Previous MMD approaches typically did not rely on pre-trained kernels, but their performances do not compare (see Table 1).
>
> > ### Missing ablation studies
> > #### 1 is addressed in the next reply, due to space constraints
> > #### 2 The paper uses hypothesis testing instead of directly using the score in equation (8). How much performance gain was obtained from this?
>
> The hypothesis testing does not improve scores (in fact, it very slightly decreases it) but is a more principled approach resulting in an unbiased estimation of the FPR. For instance, it is technically possible, for some method and for some threshold tau on the score, to achieve a FPR of 0% and a TPR of 100%. Using the framework of hypothesis testing, the p-value is lower bounded by $\frac{1}{1+|X_{val}^{(2)}|}$, which is also the p-value of the validation sample with the lowest score. Thus, the best achievable FPR for a TPR>0 is $\frac{1}{|X_{val}^{(2)}|}$. While the decrease in performance becomes negligible for a large $|X_{val}^{(2)}|$, the unbiased estimation of the FPR in the hypothesis testing framework is more rigorous.
> Additionally, this framework makes practical threshold calibration easier: one can set the mean FPR to p by setting a threshold p on the p-value (with a variance induced by the sampling of validation data), instead of having to make a line-search on the score threshold tau to achieve the desired empirical FPR (which also suffers from similar variance while additionally being biased). Finally, if the range and granularity of tau used to compute the AUROC is not adequate, the estimation of the AUROC can become very unreliable (e.g. if few points have FPR and TPR > 0). This problem does not occur with the hypothesis testing framework.
>
> > ### (Minor point) The method may be sensitive to different transformations
>
> That is an excellent point insufficiently discussed in our paper, which we propose to improve with a short sentence. We kept test transformations as simple as possible by only keeping random crops with fixed scale and horizontal flip in order to avoid sensibility to the types of transformations, and to not overfit target distributional shifts. Performances could likely be improved by tuning test transformations to each detection task, but such results would not be generalizable.

---

> > ### Author Response · Authors · 2022-11-09
> > **Author's answer 2/2**
> >
> > > ### Missing ablation studies
> > > #### 1 Computing calibration parameter gamma in Eq (7) costs n_trs * n_trs * |X_val1| * ||X_val2| computations. How much is this calibration process important? What is the performance gain from this process? What if gamma=1 is used without any calibration process?
> >
> > Similarly to the computations for MMD-CC above, while the computation complexity might seem high, it is actually very reasonable in practice. Indeed, the main cost comes from computing the embeddings for each sample and each transformation of $X_{val}^{(1)}$ and $X_{val}^{(2)}$, ie $O(n_{trs} (|X_{val}^{(1)}| + |X_{val}^{(2)}|) T_{fp})$, where $T_{fp}$ is the complexity of a single forward pass. With $|X_{val}^{(1)}|=300$, $|X_{val}^{(2)}| = 2000$ and $n_{trs} = 50$ as in our experiments, that requires 115,000 forward passes which can be done in minutes on a single A100 GPU. Once the embeddings are computed, computing $m_{out}(x)$ for each x in $X_{val}^{(2)}$ is equivalent to multiplying a matrix of shape $(100000, d_{emb})$ to a matrix of shape $(d_{emb}, 15000)$. With embeddings of dimension 2048, it can be done in less than a minute even on CPU (doing the computation sequentially to avoid memory issues). Therefore, the computational cost of the calibration step is quite negligible in practice.
> >
> > The strength of our calibration step is that it only requires clean samples, and does not require calibration on OOD samples. Most existing works either calibrate their scores on labeled OOD samples, which is sure to overfit specific distributional shifts. Some works like CSI do not require such calibration but construct their scores for a specific type of shift (see discussion at the end of section 2). The only existing work that we are aware of that calibrates their scores in the same conditions as we do is “ViM: Out-of-Distribution With Virtual Logits Matching”, where they make a similar argument regarding the importance of a calibration process agnostic to OOD samples, but they only evaluate on a single type of OOD: unseen classes.
> >
> > The capability of our approach to perform detection despite this difficult calibration setting is actually a major feat, which we have not sufficiently highlighted and on which we propose to elaborate in the “calibration” paragraph of section 5. The difficulty of this setting is highlighted in Table 4: when calibrating “Hu et al.” using the same method as we do (instead of calibrating on OOD samples), their performances drop significantly.
> >
> > Our performances could be significantly improved by tuning gamma on OOD samples, but this would undermine our argument that our method is applicable to a large variety of distributional shifts without prior knowledge of OOD samples, therefore we chose not to report such unreliable numbers. As for setting an arbitrary value to gamma, such as gamma=1, the corresponding results would similarly be arbitrary, and may perform well for a given backbone and in-distribution without transferring well to other models and datasets. Our calibration method thus leads to more reliable results than the majority of existing works, and we argue that this setting, although difficult, should be considered standard.
> >
> > > ### Clarity
> >
> > 1- $S_P^{(1)}$, $S_P^{(2)}$ and $S_Q$ all have the same size, which is a requirement for the results to be a valid p-value. We will clarify this point in the paragraph preceding section 4.1.
> >
> > 2- We repeat the experiment a 100 times, each time resampling entirely $S_P^{(1)}$, $S_P^{(2)}$ and $S_Q$, while enforcing $S_P^{(1)}$ and $S_P^{(2)}$ are disjoints.
> >
> > 3- The learned similarity indeed denotes the composition of cosine similarity with learned feature representation, i.e. the function that takes as input two images and returns the cosine similarity between their features.

---

### Decision · Program_Chairs · 2023-01-20

**Decision:**

Reject

**Justification For Why Not Higher Score:**

While the paper is quite interesting, there are multiple issues that should be addressed before being ready for publication as outlined in the latter portion of the meta-review.

**Justification For Why Not Lower Score:**

N/A

**Metareview: Summary, Strengths And Weaknesses:**

The paper proposes using self-supervised contrastive learning to detect OOD examples based on an MMD-inspired two-sample test.

The reviewers agree that the paper tackles an important problem with its new adaptation of the classical MMD to OOD detection.

However, the reviewers also point out some areas of improvement for the paper, including a more comprehensive and convincing comparison between supervised and contrastive-based approaches. Moreover, reviewers would have liked to see a better motivation for certain design choices of the method. Finally, the computational costs of the proposed method may limit its practical value.